# Quantification of Nervonic Acid in Human Milk in the First 30 Days of Lactation: Influence of Lactation Stages and Comparison with Infant Formulae

**DOI:** 10.3390/nu11081892

**Published:** 2019-08-14

**Authors:** Jiahui Yu, Tinglan Yuan, Xinghe Zhang, Qingzhe Jin, Wei Wei, Xingguo Wang

**Affiliations:** 1State Key Lab of Food Science and Technology, Jiangnan University, Wuxi 214122, China; 2International Joint Research Laboratory for Lipid Nutrition and Safety, Collaborative Innovation Center of Food Safety and Quality Control in Jiangsu Province, School of Food Science and Technology, Jiangnan University, Wuxi 214122, China

**Keywords:** n-9 fatty acid, nervonic acid, human milk fat, infant formula, lactational stage

## Abstract

Nervonic acid (24:1 n-9, NA) plays a crucial role in the development of white matter, and it occurs naturally in human milk. This study aims to quantify NA in human milk at different lactation stages and compare it with the NA measured in infant formulae. With this information, optimal nutritional interventions for infants, especially newborns, can be determined. In this study, an absolute detection method that uses experimentally derived standard curves and methyl tricosanoate as the internal standard was developed to quantitively analyze NA concentration. The method was applied to the analysis of 224 human milk samples, which were collected over a period of 3–30 days postpartum from eight healthy Chinese mothers. The results show that the NA concentration was highest in colostrum (0.76 ± 0.23 mg/g fat) and significantly decreased (*p* < 0.001) in mature milk (0.20 ± 0.03 mg/g fat). During the first 10 days of lactation, the change in NA concentration was the most pronounced, decreasing by about 65%. Next, the NA contents in 181 commercial infant formulae from the Chinese market were compared. The NA content in most formulae was <16% of that found in colostrum and less than that found in mature human milk (*p* < 0.05). No significant difference (*p* > 0.05) was observed among NA content in formulae with different fat sources. Special attention was given to the variety of n-9 fatty acids in human milk during lactation, and the results indicated that interindividual variation in NA content may be primarily due to endogenous factors, with less influence from the maternal diet.

## 1. Introduction

Human milk is generally regarded as the best source of nutrition for infants during their first six months of life [1]. Human milk contains 3%–5% fats, of which 98%–99% are triacylglycerols (TAGs), 0.26%–0.80% are phospholipids (PLs), and 0.25%–0.34% are sterols, as well as trace amounts of minor components, including monoacylglycerols (MAGs), diacylglycerols (DAGs), non-esterified fatty acids, and other substances [2]. Human milk fat is one of the most complex natural lipids with a unique fatty acids (FAs) composition. Nervonic acid (24:1 n-9, NA) is a very-long-chain monounsaturated fatty acid (>C_20_) that naturally occurs in human milk (less than 2%) [3], but it has been rarely studied in the field of human nutrition.

NA is an essential constituent of the neuronal membrane [4] and plays a crucial role in problems such as early myelination [5], peroxisomal disorders [6,7], and undernourishment [8]. NA rapidly accumulates in the fetal brain at 32–37 weeks’ gestation [9]. After birth, infants continue to obtain NA from human milk. Several publications have indicated that the NA concentration in human milk markedly decreases as lactation stages progress [10,11,12]. NA concentrations in colostrum (≤1 week postpartum) have been found to range from two- to six-fold higher than those in mature milk (>2 weeks and ≤16 weeks) [13,14]. The results suggested that the human brain heavily accumulates NA in the early days of life. Previous research identified that NA is an important fatty acid in the white matter and its deficiency in early development may damage the white matter [15]. The cause of the majority of preterm infants with cerebral palsy is mainly attributed to white matter diseases, such as periventricular leukomalacia [16,17].

The NA concentration in human milk is not routinely reported because NA quantitation is obscured by more abundant very-long-chain fatty acids when expressed percentage by weight of total fatty acids in gas chromatography (GC) analyses [18]. However, previous studies report that NA is detectable in human milk from different countries of the world, and a significantly higher NA content is present in colostrum compared to mature milk [11,12,13,19,20,21]. Very few data are available on the quantification of nervonic acid in human milk especially the amount in each lactation day.

The NA content in common vegetable oils is low, therefore infant formulae have much lower NA content compared with human milk [3], especially colostrum. However, relatively little attention has been paid to NA in infant formulae [22]. NA, together with docosahexaenoic acid (22:6 n-3, DHA) and arachidonic acid (20:4 n-6, AA), has a positive effect on the neural development of the neonate [23]. Nonetheless, the importance of DHA and AA in infant formulae have attracted researchers’ attention [9,24,25], whereas few studies focus on the effect of NA in human milk and the comparison with infant formula which is potentially important.

This study aimed to develop an effective method for the quantification of NA concentration in human milk and systematically compare the NA concentration in human milk on different lactation days. The NA contents in infant formulae from three fat sources (cows’ milk fat, goats’ milk fat, and plant oil) were also studied. Special attention was given to the variation of three other n-9 FAs, including oleic acid (18:1 n-9, OA), eicosenoic acid (20:1 n-9, EiA) and erucic acid (22:1 n-9, EA), in human milk throughout lactation.

## 2. Materials and Methods

### 2.1. Standards and Chemicals

A standard mixture of 37 kinds of fatty acid methyl esters (FAMEs, from C_4_ to C_24_) was bought from Sigma-Aldrich (St. Louis, MO, USA). The standards methyl *cis*-15-tetracosenoate (24:1 n-9 FAME) and methyl tricosanoate (23:0 FAME), methanol, and *n*-hexane (HPLC grade) were obtained from J and K Scientific (Beijing, China). Other reagents were analytical grade and purchased from Sinopharm Reagent Co. Ltd. (Shanghai, China).

### 2.2. Human Milk Samples and Infant Formulae

Human milk samples were collected from eight healthy mothers of term infants in Wuxi, China. The study was approved by the Ethics Committee of the School of Food Science and Technology in Jiangnan University (JN No. 21212030120), and written consent was obtained from all subjects included in this study. During the lactation period from 3 to 30 days postpartum, 224 human milk samples were collected. The sample-set included colostrum (3–6 postpartum days, n = 32), transitional milk (7–14 postpartum days, n = 64), and mature milk (>15 postpartum days, n = 128). All participating mothers were non-smokers, non-medicated, and healthy. The mean age of the mothers was 28.38 ± 3.16 years, mean BMI was 21.78 ± 2.75 kg/m^2^, and mean infant weight at birth was 3.35 ± 0.23 kg. The samples (5–10 mL) were collected after full expression from one breast with a breast pump between 9 and 11 am. The samples were stored at −20 °C for less than 2 h and transferred to the lab at Jiangnan University for lipid extraction.

The fatty acid composition of infant formulae is reported in a previous study [3]. A total of 181 formulae from 27 brands were collected; the tested formulae account for 75% of the Chinese infant formulae market. Three stages (infant, follow-on, and growing-up) of formulae were included. The formulae were divided into cows’ milk, goats’ milk, and plant oil, depending on the main fat source.

### 2.3. Fatty Acid Methyl Esters Preparation

Total lipids in human milk were extracted by classic Röse–Gottlieb method [26] with some modifications. Briefly, 1 mL ammonia water was added into 4 mL human milk, and then mixed in a water shaking bath at 65 °C ± 5 °C for 20 min. Then, 5 mL absolute ethanol, 10 mL absolute ether and 10 mL petroleum ether were added to extract the lipids. The samples were mixed thoroughly and stood for 2 h, and then the supernatants were collected. The lipids in the lower phase were extracted using half of the solvents as above. The solvents were removed by nitrogen and the lipids were stored in a −80 °C freezer until analysis.

The 24:1 n-9 FAME standard was prepared at a concentration of 0.5 mg/mL dissolved in *n*-hexane. The internal standard solution was prepared by weighing 10 mg of 23:0 FAME into 10 mL of *n*-hexane. Milk fat (40 mg) was suspended in 200 μL of the internal standard solution, 800 μL *n*-hexane, and 500 μL of KOH–CH_3_OH (2 mol/L). After mixing for 2 min, the water in the solution was removed by adding the appropriate amount of anhydrous sodium sulfate. Then, the mixture was mixed thoroughly and left standing for half an hour. The supernatant was filtered by a 0.22 μm filter and analyzed by gas chromatography (GC).

### 2.4. GC Analysis

The samples were analyzed by an Agilent 7820A gas chromatograph (Agilent Technologies, Santa Clara, CA, USA) with a hydrogen flame ionization detector, and they were separated by a TRACE^TM^-FAME capillary column (60 m × 0.25 mm × 0.25 μm, Thermo Fisher Scientific, Waltham, MA, USA) using nitrogen carrier gas [27]. The injection and detector temperature were set at 250 °C, and the injection volume was 2 μL. The temperature of the column was set at 60 °C for 3 min to start and then increased to 175 °C at a rate of 5 °C/min. The temperature was maintained at 175 °C for 15 min and then raised (at a rate of 2 °C/min) to 220 °C, which was maintained for 10 min. FAs were identified by comparing the retention times of the sample peaks with those of the FAME standards. The NA concentration was measured by the absolute detection method, with 23:0 FAME as an internal standard.

### 2.5. Quantitation of Nervonic Acid

In the first step, a series of standard solutions that contained varying concentrations of 24:1 n-9 FAME and an identical amount of 23:0 FAME were injected. The calibration line was constructed using Equation (1):(1)y=ax+b
where *y* is the ratio of the peak area of 24:1 n-9 FAME to that of 23:0 FAME, a is the slope of the calibration curve, *x* is the ratio of the weight of 24:1 n-9 FAME to that of 23:0 FAME and b is the intercept of the calibration curve.

The NA concentration in human milk fat was calculated using Equation (2), with 23:0 FAME as the internal standard.
(2)$$C=\left(\frac{A_{1}}{A_{0}}-b\right)\times m_{0}\times \frac{1}{a}\times \frac{1}{W}\times F_{t}$$
where C is the concentration of NA in human milk fat (mg/g fat), *A*_1_ is the peak area of 24:1 n-9 FAME, *A*_0_ is the peak area of 23:0 FAME, *m*_0_ is the weight of the internal standard/mg, W is the weight of human milk fat in the test sample/g, and *F_t_* is the transformation coefficient of 24:1 n-9 FAME to NA.

### 2.6. Validation of the Method

The following parameters (precision, recovery rate, and limits of detection and quantification) were used to validate this method based on the guidance of the United States Pharmacopeia (USP) [28]. For the precision test of the method, the same homogeneous sample was analyzed three times in one day to obtain the intra-day precision and in three different days to obtain the inter-day precision. The precision was described as peak area and retention time relative standard deviations (RSDs) [29]. The recovery of the methods was measured using three levels of concentration of standard solution that were added into three identical human milk samples, and no standard was added into the fourth sample. The rate was calculated by the detected concentration of NA standard divided by the actually added concentration. The limits of detection and quantitation were defined as three times and ten times of signal to noise ratio, respectively [30].

### 2.7. Statistical Analysis

All analyses of human milk were performed in duplicate. The results were expressed as means (%) ± standard deviations (SD) and were calculated using the SPSS 19.0 (SPSS, USA). Differences in NA concentration were tested by one-way analysis of variance (ANOVA) for continuous variables. Two-way ANOVA was used to evaluate the effect of fat sources and stages on the n-9 FA composition of infant formulae. Differences among all results were compared by use of Ducan’s test at *p* < 0.05. Pearson’s correlation test was used to determine the correlation coefficient between NA concentration and lactation days. Principal component analysis (PCA) was used to determine the differences in NA content in three lactation stages of human milk and three stages of infant formulae using SIMCA-P software version 13.5 (Demo Umetrics, Umea, Sweden).

## 3. Results

### 3.1. Nervonic Acid Concentration in Human Milk

In this study, to accurately quantify the NA concentration in human milk, a long GC program with an analysis time of 70 min was applied, and a FAME with a similar molecular structure (23:0) was added to the sample as an internal standard [31]. The gas chromatograms of NA in human milk are shown in Figure 1.

The calibration line of this method was defined by y = 0.733x + 0.7417 (R^2^ = 0.9944) showing a good linearity [32]. The calibration lines of 24:1 n-9 and 23:0 were y = 9.47 × 10^6^x (R^2^ = 0.9938) and y = 9.59 × 10^6^x (R^2^ = 0.9965), respectively. The precision measured the dispersion degree between the tested results of the same sample under the same condition [29]. Both the intra-day and inter-day RSD% on the retention times were lower than 0.01%, and the intra-day and inter-day RSD% on the peak areas were lower than 0.02% and 0.18%, respectively. The recovery rate of the method ranged from 95% to 103%, which met the advisable international level of 80% to 115% [33]. The detection limits ranged from 1.34 μg/mL for NA to 3.25 μg/mL for 23:0 FAME and the quantitation limits ranged from 3.11 μg/mL to 5.36 μg/mL for NA and 23:0 FAME, respectively.

The NA concentrations in human milk are presented in Appendix A. The NA concentration in all human milk samples decreased significantly (r = −0.822, *p* < 0.001) during the first month of lactation. The average NA concentration on day three of lactation was about five times higher than that on day 30 (*p* < 0.001), with values of 1.00 ± 0.24 mg/g and 0.18 ± 0.03 mg/g fat, respectively. No significant difference was observed in the NA concentration in human milk obtained after 15 days of lactation. The concentration of NA was highest in colostrum (0.76 ± 0.23 mg/g fat) and lowest in mature milk (0.20 ± 0.03 mg/g fat), decreasing by 82% in the first month of lactation. Figure 2 illustrates the changing trend of the mean values of NA concentration in human milk in the first month of lactation.

Besides changes between lactation stages, there were significant interindividual differences in NA concentration. The differences were more pronounced from three to 15 days and marginally significant (*p* > 0.05) after day 15 of lactation. At three days postpartum, there was a two-fold variation in NA in the milk from the eight mothers, with values ranging from 0.72 to 1.44 mg/g fat. The differences between individuals decreased over time during the study period. During days 26–30 of lactation, the NA concentration varied from 0.15 mg/g fat to 0.22 mg/g fat, with a 0.07 mg/g fat difference.

### 3.2. Composition of n-9 Fatty Acids in Human Milk

In this study, four kinds of n-9 FAs (OA, EiA, EA, and NA) were detected in human milk samples. Changes in n-9 FA composition in human milk during the first month of lactation are shown in Figure 3. OA, whose percentage ranged from 30.23% to 33.86% of total FAs, was the predominant FA in human milk. During lactation, no significant difference was observed in OA content in human milk (Figure 3A). Contrary to the observed consistency of OA, there were significant changes (*p* < 0.05) in the percentage of EiA (from 0.17% to 0.06%) and NA (from 0.21% to 0.04%) as the milk progressed from colostrum to transitional milk and from transitional milk to mature milk (Figure 3b,d). Particularly, the NA percentage varied in the range of 0.18%–0.21% in colostrum and decreased to 0.12%–0.17% in transitional milk and 0.04%–0.10% in mature milk. The EA content decreased from 0.19% to 0.11% of the total FA content from the first to third lactation stage, but the discrepancies were less significant between transitional milk and mature milk (Figure 3c).

### 3.3. Composition of n-9 Fatty Acids in Infant Formulae

A total of 181 infant formulae were analyzed, and 97 (53.59%) were found to contain NA. The n-9 FA composition of 97 infant formulae are presented in Table 1. The infant formulae were divided into three stages (infant formula, follow-on formula, and growing-up formula) and three sources (cows’ milk formula, goats’ milk formula, and plant-oil formula). There was a significant difference in fat sources of OA (*p* < 0.001), EiA (*p* < 0.001), and EA (*p* < 0.01) among different formulae. There were no significant differences in sources of NA (*p* > 0.05). The n-9 FA composition was not affected by the formulae stages or the interaction between fat sources and stages (*p* > 0.05). It is recommended that EA content in infant formulae be below 1% of the total fat content [34]. It is noteworthy that infant formulae have much lower concentrations of NA compared with human milk. The NA content of most formulae (0.02% ± 0.02%) was about <16% of that in colostrum (0.20% ± 0.04%) and less than from that found in mature milk (0.08% ± 0.04%).

## 4. Discussion

NA occurs naturally in human milk; however, it has not been routinely reported in the global FA profile of human milk [18]. The main reason for its frequent omission is that the concentration of NA in human milk is low, and its retention time is similar to several polyunsaturated fatty acids (PUFAs). These factors make the quantitation of NA very difficult. In this study, 23:0 FAME was used as an internal standard to quantify NA concentration in human milk fat. In general, the validation of the method for the quantification of NA in human milk indicated that the method parameters’ were well within internationally accepted limits.

The NA concentration significantly decreased (*p* < 0.05) with the progression of lactation days. Furthermore, the NA level decreased markedly during the first 10 days then decreased slowly after 15 days. The fat contents of human milk increased during lactation (Appendix A). As shown in Table 2, the NA contents in human milk from mothers in different geographical regions and at different lactation stages were summarized. Most of the studies show a significantly higher NA content in colostrum compared to mature milk [11,12,13,19,20]. This study indicates the concentration of NA in colostrum was three times higher than that in mature milk, which is similar to Hua et al. [19] and Rueda et al. [20]. The results of the NA concentration in human milk during the three stages (colostrum, transitional milk, and mature milk), were similar to those found by Xiang [35]. Other publications show ten times higher NA content in colostrum compared to mature milk [11,12]. However, there are also a few publications that report the NA concentrations decrease from 0.35 to 0.27 mg/g of human milk, with no significant difference between colostrum and mature milk [36].

There are some discrepancies between studies, as shown in Table 2. The percentage of NA in human milk from individuals in Wuxi ranged from 0.06% to 0.20%. This result is marginally different from the values from several previous studies, which have reported values ranging from 0.19% to 0.99%. The difference could be due to analytical methods, as well as individuals from different locations and the lactation day. Further studies on the absolute detection of NA concentration from different countries and the correlation with maternal diet are needed.

Dietary habits differ among regions and can cause internal physiological differences [44]. Jing and co-workers determined the NA content of transitional milk and mature milk from five regions (Shanghai, Guangzhou, Nanchang, Harbin, and Hohhot) of China, and the content in Hohhot was highest, regardless of whether the sample was transitional milk or mature milk [40]. Diets containing NA were fed to lactating rats, and the NA in the diet influenced the NA concentration in the animal’s milk [45]. Although it is rare for NA to be provided by a typical diet, earlier studies have reported that dietary OA and EA influence the nervonic acid content in humans [46]. Lactating women are able to synthesize NA by carbon chain elongation with enzyme catalysts [47]. The large variation in NA concentration in the milk of human mothers from different regions may reflect differences in the mothers’ diets [48].

There are differences in the diets of lactating mothers around the world that can account for the NA concentration differences between individuals. Additionally, the individual variation may also be related to maternal age [49], and Body Mass Index (BMI) [50]. However, in this study and previous studies, the NA concentration in all samples had a downward trend throughout the entire lactation period. This consistency, despite regional differences, indicates that the decline in NA content in human milk is more likely to be influenced by an endogenous factor [51] than a dietary habit.

The amounts of DHA and AA in human milk, similarly to NA, had a decreasing trend over the lactation period (Appendix A). This has been attributed to the genes that encode delta (5)- and delta (6)-desaturases and their effect on the proportions of these FAs in human milk [52]. The DHA and AA precursors linolenic acid (18:3 n-3, ALA) and linoleic acid (18:2 n-6, LA) increase in parallel [53]. The hypothetical pathways of n-9 FAs metabolism are shown in Figure 4. NA is synthesized from OA by elongation. The OA content measured in this study (30.23%–33.86%) was in accordance with Qi’s results (30.66%–32.71%) [27], but inconsistent with Weiss’s study (43.96%–48.21%) [53]. OA content was stable in this study, which was contrary to the study by Lopez-Lopez et al., who claimed that OA content decreased as lactation progressed [54]. Additionally, the percentage of EiA, EA, and NA decreased with the progression of lactation, which agreed with the results obtained by Nyuar et al. [11]. In general, the amount of OA was stable over time, and the other precursors, EiA and EA, decreased during the same period, which differs from the precursors of DHA or AA. The mechanism by which genes regulate the metabolism of n-9 FAs remains unclear and needs further study. In human milk, OA is the richest FA and serves as a structural component. It can be assumed that the OA content might be sufficient to synthesize NA and other n-9 FAs.

It has been demonstrated that infants have the capacity to convert ALA and LA to DHA and AA, respectively, and that elongases and desaturases have activity in the first week after birth [55]. However, the ability of infants to synthesis FAs is weak, and the amount of DHA formed from ALA is inadequate to support brain development [56]. For the optimal development of an infant’s brain, a sufficient amount of NA, DHA, and AA is necessary for newborns.

Human milk is a rich source of NA compared with commercial infant formulae. In the 118 collected infant formulae samples, only 53.59% contained NA. The PCA revealed differences of NA content between human milk (including colostrum, transitional milk, and mature milk) and infant formulae (Figure 5). A significant distinction of NA content was observed between human milk and infant formula (*p* < 0.05). NA content in the three-stages (colostrum, transitional milk, and mature milk) of human milk differs obviously (*p* < 0.05). However, there is no significant difference of NA content in the three types (infant, follow-on, growing-up) of infant formula. Most NA-containing formulae had NA concentrations that were <16% of that measured in colostrum. NA content in formulae needs to be enhanced urgently for term newborns and especially for preterm newborns. It has been reported that the NA concentration in preterm milk is seven-fold higher compared to that in milk from mothers with full-term babies at a similar stage of lactation [9].

The NA level is considered to reflect brain maturation, and its accumulation in the brain is a sign of the onset of myelinogenesis [57]. Newborns who cannot be breastfed fail to get sufficient levels of NA from infant formulae. Long-term NA deficiency will hamper the development of the nervous system and cause visual impairment [58]. Supplying dietary lipids to stimulate the synthesis of n-9 FAs has been advised to support myelination in newborns [23]. Thus, infant formulae need to enhance their NA content to match the NA content in human milk. This applies to formulae for full-term infants, and it is especially the case for preterm infants.

NA content in formulae was not related to fat sources or stages. However, the differences observed between other n-9 FAs in infant formulae were particularly influenced by fat sources. The current formula fat sources (generally used cow milk fat, goat milk fat, and vegetable oils) contain no or trace amount of NA. NA has been found in some plant species [59] for example *Malania oleifera* [60] and *Lunaria annua* [61]. However, oils of these plants are rarely developed and used in infant formulae because of their high EA content. It has been demonstrated that the heart may be damaged by EA, and EA is undesirable for human consumption [62]. It is necessary to explore the effect of NA as well as different forms (FA, TAG or PLs) for the optimal brain development of infants.

## 5. Conclusions

The NA concentration was at its highest at the onset of lactation, with an overall decrease of 82% throughout the 3–30 days postpartum study period. The concentration dropped at a faster rate in the first 10 days and then flattened out. Thus, it is crucial for infants, especially preterm infants, to get enough NA within the first ten days of lactation. The NA concentration differs between breastfeeding mothers, in part because of the regions in which they live. Further studies are needed to clarify the NA biosynthetic system in infants. Most of the 97 infant formulae in which NA was detected had less than 16% of the NA content in colostrum. There were significant differences between the fat sources of OA, EiA, and EA in the formulae, but this was not the case for NA. The effect of the stages or the interaction between sources and stages on infant formulae FA composition was not obvious. Because of the low level of NA found in infant formulae, it is suggested that the NA content in infant formulae for newborns be increased. Further work is also warranted to identify the best form of NA for infants.

## Figures and Tables

**Figure 1 nutrients-11-01892-f001:**
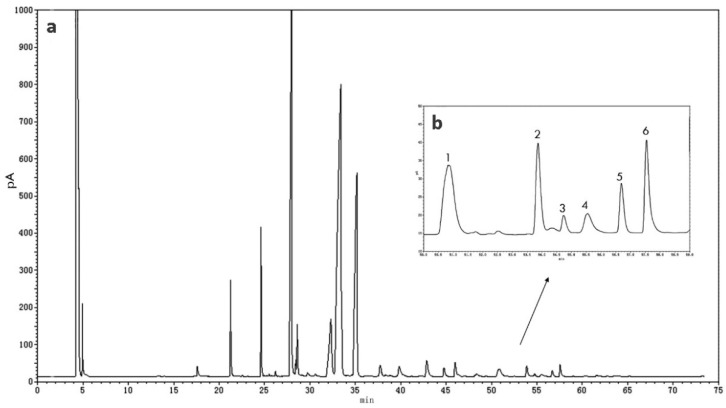
Gas chromatography (GC) of total fatty acids (**a**) and very-long-chain fatty acids (**b**) in one colostrum human milk. 1, tricosanoicacid (23:0); 2, docosatetraenoic acid (22:4 n-7); 3, docosapentaenoic acid (22:5 n-6); 4, nervonic acid (24:1 n-9, NA); 5, docosapentaenoic acid (22:5 n-3); 6, docosahexaenoic acid (22:6 n-3).

**Figure 2 nutrients-11-01892-f002:**
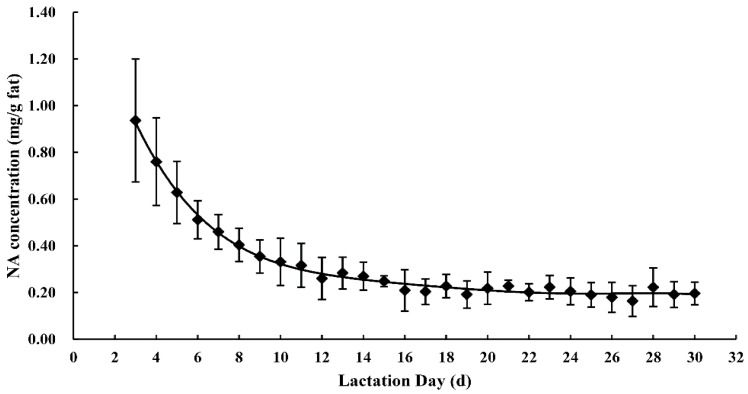
Nervonic acid concentrations in human milk throughout the first month of lactation.

**Figure 3 nutrients-11-01892-f003:**
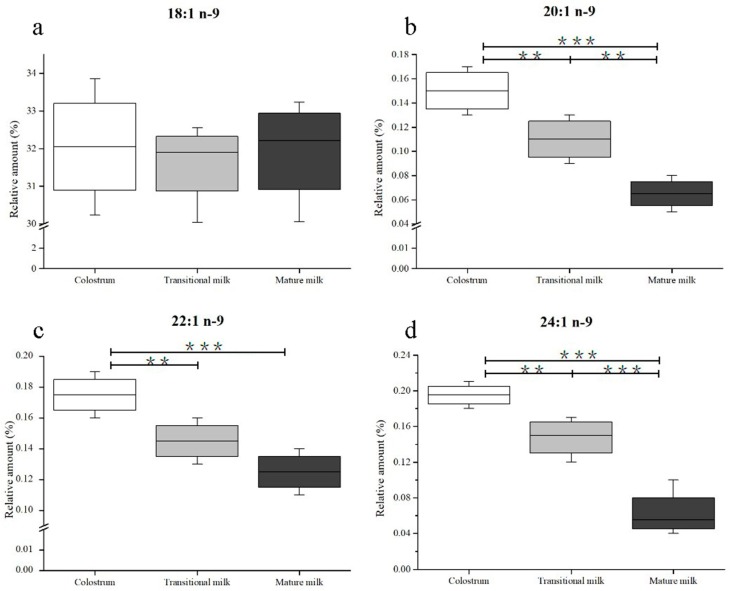
Changes in n-9 fatty acids including 18:1 n-9 (**a**), 20:1 n-9 (**b**), 22:1 n-9 (**c**), and 24:1 n-9 (**d**) in human milk during the first month of lactation. Significantly different from colostrum, transitional milk, and mature milk: *** *p* < 0.001; ** *p* < 0.01. No labeling indicates no significant differences.

**Figure 4 nutrients-11-01892-f004:**
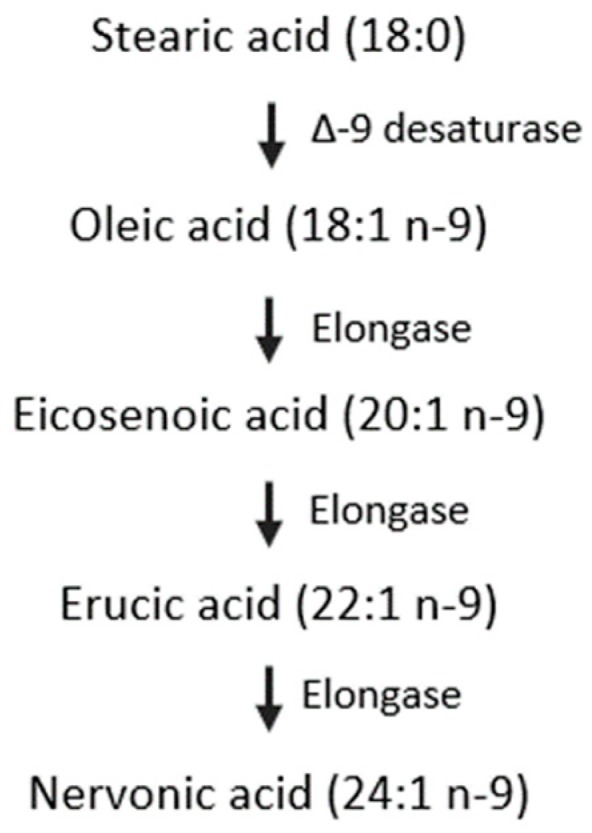
Hypothetical pathway for the synthesis of nervonic acid in infants.

**Figure 5 nutrients-11-01892-f005:**
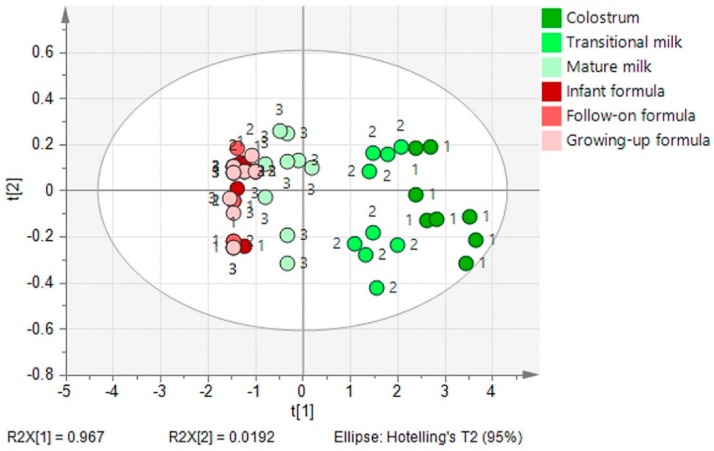
Principal component analysis of stage difference in the NA content in human milk (1, colostrum; 2, transitional milk; 3, mature milk) and infant formulae (1, infant formula; 2, follow-on formula; 3, growing-up formula).

**Table 1 nutrients-11-01892-t001:** n-9 Fatty acid composition (wt%) of infant formulae.

Fatty Acids		CMF	GMF	POF	*p*-Value		
		(n = 30)	(n = 16)	(n = 51)	F	S	F × S
18:1 n-9	IF	32.15 ± 5.06 ^a^	34.81 ± 5.66 ^a,b^	40.74 ± 8.32 ^b^	***	NS	NS
	FF	30.65 ± 5.95 ^a^	30.15 ± 5.28 ^a^	41.61 ± 6.11 ^b^			
	GF	31.76 ± 5.09 ^a^	29.66 ± 6.50 ^a^	40.89 ± 7.31 ^b^			
20:1 n-9	IF	0.23 ± 0.09	0.29 ± 0.09	0.31 ± 0.10	***	NS	NS
	FF	0.17 ± 0.10 ^a^	0.24 ± 0.09 ^a,b^	0.33 ± 0.09 ^b^			
	GF	0.13 ± 0.04 ^a^	0.21 ± 0.07 ^a^	0.30 ± 0.09 ^b^			
22:1 n-9	IF	0.04 ± 0.04	0.06 ± 0.04	0.04 ± 0.03	**	NS	NS
	FF	0.03 ± 0.03	0.05 ± 0.03	0.05 ± 0.03			
	GF	0.01 ± 0.01 ^a^	0.04 ± 0.03 ^a,b^	0.05 ± 0.04 ^b^			
24:1 n-9	IF	0.03 ± 0.13	0.03 ± 0.02	0.03 ± 0.02	NS	NS	NS
	FF	0.02 ± 0.00 ^a^	0.02 ± 0.01 ^a^	0.03 ± 0.01 ^b^			
	GF	0.02 ± 0.01	0.02 ± 0.01	0.03 ± 0.01			

CMF, cows’ milk formula; GMF, goats’ milk formula; POF, plant-oil formula; IF, infant formula; FF, follow-on formula; GF, growing-up formula. F, fat source; S, stage. Different superscript lowercase letters indicate significant differences (*p* < 0.05) with a row. *** *p* < 0.001; ** *p* < 0.01. NS, *p* > 0.05.

**Table 2 nutrients-11-01892-t002:** Mean value of nervonic acid (wt%) in human milk from individuals in different regions.

Regions	Colostrum	Transitional Milk	Mature Milk	References
Wuxi, China	0.20 (3–6 d)	0.15 (7–14 d)	0.06 (15–30 d)	This study
Taiwan, China	0.99 (1–6 d)	-	0.28 (2 m)	Hua et al. [19]
Switzerland	0.39 (1 week)	0.13 (2 weeks)	0.07 (3–8 weeks)	Thakkar et al. [13]
Beijing, China	0.54 (4 d)	-	0.25 (30 d)	Zhao et al. [14]
Thailand	-	-	0.06	Golfetto et al. [37]
Korean	-	0.27	-	
Bangladeshi	-	-	0.20	Yakes et al. [38]
Northern Sudanese	0.19	0.15	0.02	Nyuar et al. [11]
Wenzhou, China	0.45	-	-	Peng et al. [39]
Changzhou, China	0.25	-	-	
Shanghai, China	-	0.08	0.05	Jing et al. [40]
Guangzhou, China	-	0.06	0.06	
Nanchang, China	-	0.12	0.11	
Harbin, China	-	0.06	0.04	
Hohhot, China	-	0.21	0.19	
Granada, Spain	0.28	0.08	0.07	Sala-Vila A et al. [21]
Congolese	-	-	0.04	Rocquelin et al. [41]
Panama	0.32	0.16	0.10	Rueda et al. [20]
Spain	0.24	0.17	0.10	
Dominica	-	-	0.05	Beusekoma et al. [42]
Saint Lucia	0.41	0.11	0.04	Boersma et al. [12]
Belize	-	-	0.06	Cheristien et al. [43]
Dominica	-	-	0.02	

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
