# Peer review of "Quantification of Nervonic Acid in Human Milk in the First 30 Days of Lactation: Influence of Lactation Stages and Comparison with Infant Formulae"

_nutrients, 2019, doi:10.3390/nu11081892_

Round 1

Reviewer 1 Report

This observational study examines nervonic acid (NA) in human milk and infant formula in China. The methods are relatively well described, and the authors make note of existing literature examining NA content in milk. This study is well written however it is unclear how the analytical method employed compares to current methods and reporting of NA in this population adds to what is currently known. Therefore, the novelty of the study in the context of the current literature and methodology is not clear and should be emphasized in this manuscript. Suggestions are revisions are included below. 

1.     Please state in the abstract and line 212 how NA content in infant formula compares not only to colostrum, but mature milk as well.

2.     In the introduction, please ensure the correct citation is used (reference 15), and state associated, rather than caused.

3.     Please provide further details on the mixture of 37 FAMEs used as a standard.

4.     From the sample size of n=8, it is unknown how this was used to formulate the values in Table 1. How many samples and timepoints were taken from each woman? And how was this variation in the number of samples considered in the statistical analysis? Moreover, how was missing data or difference in sample size for each of the lactation days considered in the statistical model?

5.     Further details are how samples were obtained are needed. Were women given specific instructions on expression on milk, and how were the samples stored and transported prior to storing at -80°C?

6.     How do the stages of infant formula (infant, follow-on and growing up) compared to lactation stages?

7.     The authors give a reference (20) for how lipids were extracted, but please briefly describe in the manuscript.

8.     Line 86 – was the 24:1 n-9 FAME standard the mixture of 37 FAMEs mentioned in 2.1?

9.     Please explain the rationale for using a 23:0 FAME as an internal standard.

10.  Explain the statistical method used in analyzing the differences between women (Table 1). Additionally, please explain the rationale for examining differences between the n=8 women. Was the sample size large enough to detect meaningful differences in NA content?

11.  Figure 1 - please add the FA names to either footnote of on the figure.

12.  Figure 2- was this a mean of NA content for each of the lactation days?

13.  Table 2- how were the n=181 infant formulae represented in this table?

14.  Line 219 – states NA decreases over days of lactation, but then states concentrations remain stable after day 15. As this is contradictory, please provide more details on the pattern of NA over the course of lactation days.

15.   The range of 0.06-0.020% in this study is noted as marginally different than previous studies reporting a range of 0.19-0.99%. However, this is more than just marginally different, and on the lower end of all other studies. The authors explain the difference could be due to analytical methods and the objectives state that the aim is to develop an effective method for quantification of NA. However, the authors fail to discuss in the introduction or discussion what the current issues are with quantifying NA are, and why this proposed method is effective.

16.  In the discussion, please include what factors may be considered in explaining the discrepancies in the NA content between populations.

17.  The abstract and discussion conclude that interindividual variation in NA content may be due to endogenous factors rather than dietary intake. However, the authors did not look at either of these factors to make this conclusion. Please revise and expand on why diet was not examined.

18.  Line 133 – abbreviated to ANOVA.

Author Response

Response to Review Comments

(Manuscript ID nutrients-552980)

Reviewers' comments:

Reviewer #1: This observational study examines nervonic acid (NA) in human milk and infant formula in China. The methods are relatively well described, and the authors make note of existing literature examining NA content in milk. This study is well written however it is unclear how the analytical method employed compares to current methods and reporting of NA in this population adds to what is currently known. Therefore, the novelty of the study in the context of the current literature and methodology is not clear and should be emphasized in this manuscript. Suggestions are revisions are included below.

The authors thank the reviewer’s comments. We have addressed the reviewer’s comments from point by point as follows. The corresponding modifications are now included in the revised version of the manuscript.

1. Please state in the abstract and line 212 how NA content in infant formula compares not only to colostrum, but mature milk as well.

The authors thank reviewer’s comments. Added as suggested (line 26).

2. In the introduction, please ensure the correct citation is used (reference 15), and state associated, rather than caused.

The authors thank reviewer’s comments. This part has been rewritten (line 50-51).

3. Please provide further details on the mixture of 37 FAMEs used as a standard.

A standard mixture of 37 kinds of fatty acid methyl esters (FAMEs) was bought from Sigma-Aldrich (St. Louis, MO, USA). The information has been added into the Materials.

4. From the sample size of n=8, it is unknown how this was used to formulate the values in Table 1. How many samples and timepoints were taken from each woman? And how was this variation in the number of samples considered in the statistical analysis? Moreover, how was missing data or difference in sample size for each of the lactation days considered in the statistical model?

This study was designed to analysis the NA and fatty acid composition in human milk during every lactation day in the first month. More than 10 volunteer mothers were enrolled. However, only 8 participants were successful exclusive breastfeeding. The human milk samples were collected from 8 participants every day from 3 to 30 days postpartum. Therefore, a total of 224 samples were obtained as shown in Table S1.

5. Further details are how samples were obtained are needed. Were women given specific instructions on expression on milk, and how were the samples stored and transported prior to storing at -80°C?

More detailed information was added to the manuscript where related (line 88-90).

6. How do the stages of infant formula (infant, follow-on and growing up) compared to lactation stages?

We have added a principal component analysis (PCA) to determine the differences in NA content in three lactation stages of human milk and three stages of infant formulae (Figure 5).

7. The authors give a reference (20) for how lipids were extracted, but please briefly describe in the manuscript.

The authors thank reviewer’s comments. Added as suggested.

8. Line 86 – was the 24:1 n-9 FAME standard the mixture of 37 FAMEs mentioned in 2.1?

They are of different standards. The mixture of 37 FAMEs contains 24:1 n-9. However, for absolute quantification of 24:1 n-9 concentration, a calibration curve was set up using 24:1 n-9 FAME standard and 13:0 FAME as an internal standard.

9. Please explain the rationale for using a 23:0 FAME as an internal standard.

The internal standard needs to be absent in human milk and has similar physicochemical properties with NA which requires that the retention time of NA and internal standard is close [1]. Based on it, 23:0 FAME was chosen as the internal standard in this study.

10. Explain the statistical method used in analyzing the differences between women (Table 1). Additionally, please explain the rationale for examining differences between the n=8 women. Was the sample size large enough to detect meaningful differences in NA content?

Differences between women were tested by one-way analysis of variance (ANOVA) for continuous variables which has been added to the manuscript. We agree to the reviewer that the sample size is small (eight mothers), but we think the final set of samples were qualified (224 samples over 3-30d postpartum). We have some conditions to filter the human milk samples, 1) the mothers are exclusive breastfeeding in at least the first month of lactation, 2) the samples were collected every day during lactation, 3) the sample volume were enough for the NA analysis and the lipids in human milk were extracted as soon as the sample collected. The samples were difficult to obtain.

11. Figure 1 - please add the FA names to either footnote of on the figure.

Added as suggested (line 172-175).

12. Figure 2- was this a mean of NA content for each of the lactation days?

Yes, this is a mean of NA content in human milk fat of the 8 mothers. We also labelled the standard deviation of NA content of the lactation days.

13. Table 2- how were the n=181 infant formulae represented in this table?

A total of 181 infant formulae were analyzed, and 97 (53.59%) were found to contain NA. The n-9 FA composition of 97 infant formulae are presented in Table 2. And the 97 infant formulae were divided into three stages (infant formula, follow-on formula and growing-up formula) and three sources (cows' milk formula, goats' milk formula and plant-oil formula). This detailed information has been added to the manuscript where related.

14. Line 219 – states NA decreases over days of lactation, but then states concentrations remain stable after day 15. As this is contradictory, please provide more details on the pattern of NA over the course of lactation days.

The concentration decreased markedly during the first ten days and then declined slowly after 15 d. From 15 to 30 day, the concentration of NA still decreased, but the trend was slow down. This part has been rewritten.

15. The range of 0.06-0.020% in this study is noted as marginally different than previous studies reporting a range of 0.19-0.99%. However, this is more than just marginally different, and on the lower end of all other studies. The authors explain the difference could be due to analytical methods and the objectives state that the aim is to develop an effective method for quantification of NA. However, the authors fail to discuss in the introduction or discussion what the current issues are with quantifying NA are, and why this proposed method is effective.

The authors thank the reviewer’s comments. Follow the reviewer’s comments, and more discussion has been added into the Introduction (line 52-28).

16. In the discussion, please include what factors may be considered in explaining the discrepancies in the NA content between populations.

Besides the dietary habits, the individual variation also may be related to maternal age [2], and BMI [3]. These have been added to the discussion (line 266-267).

17. The abstract and discussion conclude that interindividual variation in NA content may be due to endogenous factors rather than dietary intake. However, the authors did not look at either of these factors to make this conclusion. Please revise and expand on why diet was not examined.

This study failed to exam the impact of maternal diet on the NA concentration of human milk. However, as far as we know, the NA concentration in the normal diet is very low. NA generally occurs in some plant oils. We previously analyzed the fatty acid composition in commonly edible vegetable oils using the same analytical method we used in this study [4]. The results indicated that  NA was not detected in all 18 edible vegetable oils. According to the publications, NA has been found in some plant species [5] for example Malania oleifera [6] and Lunaria annua [7], which are very uncommon edible oils. However, the NA has been detected in all human milk samples, and the same decreased trend were observed. Therefore, we raised a hypothesis that NA in human milk may be influenced by the endogenous factors. Further study is needed.

18. Line 133 – abbreviated to ANOVA.

Corrected as suggested.

References

1.             Analysis, I.W.G.o.M.o. Guidelines for the quantitative gas chromatography of volatile flavouring substances, from the Working Group on Methods of Analysis of the International Organization of the Flavor Industry (IOFI). Flavour & Fragrance Journal 2011, 26, 297-299.

2.             Argov-Argaman, N.; Mandel, D.; Lubetzky, R.; Kedem, M.H.; Cohen, B.C.; Berkovitz, Z.; Reifen, R. Human milk fatty acids composition is affected by maternal age. J. Matern.-Fetal Neonatal Med. 2017, 30, 34-37, doi:10.3109/14767058.2016.1140142.

3.             Marín, M.C.; Sanjurjo, A.; Rodrigo, M.A.; de Alaniz, M.J. Long-chain polyunsaturated fatty acids in breast milk in La Plata, Argentina: relationship with maternal nutritional status. Prostaglandins Leukotrienes & Essential Fatty Acids 2005, 73, 355-360.

4.             Wei, W.; Sun, C.; Jiang, W.; Zhang, X.; Hong, Y.; Jin, Q.; Tao, G.; Wang, X.; Yang, Z. Triacylglycerols fingerprint of edible vegetable oils by ultra-performance liquid chromatography-Q-ToF-MS. LWT 2019, 112, 108261, doi:https://doi.org/10.1016/j.lwt.2019.108261.

5.             Fan, Y.; Meng, H.-M.; Hu, G.-R.; Li, F. Biosynthesis of nervonic acid and perspectives for its production by microalgae and other microorganisms. Appl. Microbiol. Biotechnol. 2018, 102, 3027-3035, doi:10.1007/s00253-018-8859-y.

6.             Wang, X.Y.; Fan, J.S.; Wang, S.Q. Development situation and outlook of nervonic acid plants in China. China Oils & Fats 2006.

7.             Yiming, G.; Elzbieta, M.; Tammy, F.; Vesna, K.; Brost, J.M.; Michael, G.; Barton, D.L.; Taylor, D.C. Increase in nervonic acid content in transformed yeast and transgenic plants by introduction of a Lunaria annua L. 3-ketoacyl-CoA synthase (KCS) gene. Plant Mol.Biol. 2009, 69, 565-575.

Reviewer 2 Report

Comments and suggestions for Authors

This paper nicely discusses the quantification of nervonic acid in huamn milk over the first month of lactation and compare with the level of different artificial milk. The manuscript covers the aim of the journal and the subject investigated is of worldwide interest. However, some points need to be addressed and the manuscript needs re-editing of some fragments.
Overall
As stated by Authors, „eight healthy Chinese mothers were included to the study”, however, there are a lot of scientific reports which highlight the interindividual variation of different factors among lactating mothers, also by the authors themselves, so in my opinion, the obtained results, due to the low numer of lactating mothers, should be reported as preliminary and this should be reflected in the title of the manuscrypt.
Was the total fat concentration determined in the analyzed milk? For example, by using Miris Milk Analyser. The results should be correlated with the total fat content. This would benefit and refine the obtained results. Moreover, I have doubts whether the calculation of statistical differences among individual mothers (at the subsequent days of lactation) is justified (see Table 1 – last column, p value). Under the Table, the details concerning the statistical test used are missing. Moreover, the detailed data in Table 1 should be presented as supplementary file.

Line 46-47
„NA concentrations in colostrum have been found to range from 2- to 6-fold higher than those in mature milk [13,14].”
It benefits the reader if the Authors will added information „how mature milk”, namely weeks or months.
Line 70-79
2.2. Human milk samples
Some very important details concerning collections of milk are missing (i.e. using an electric breast pump, sampling of a single breast, etc.).
The detailed description is needed since there are a lot of data which point out the significant differences in human milk lipid fraction during single feeding (fat levels changing foremilk to hindmilk) as well as over the day.
Line 111
2.5. Quantitation of nervonic acid
„The NA concentration in human milk fat…” – the method used by Authors is semi-quantitative, so it would be more appropriate to use the word „level”.

Results

Some subsections contain the sentences which shoud be transferred to the Discussion. I suggest that the authors should modified these fragments:
Line 138-144
„NA occurs naturally in human milk; however, it has not been routinely reported in the global FA profile of human milk [24]. The main reason…”
Line 152-155
„The limits of detection and quantitation were lower than previous studies by Barros et al. [28] and Arcari et al. [29]. In general, …”
Line 164-166
„The results of the NA concentration in human milk during the three stages, that is, colostrum, transitional milk, and mature milk, are similar to those found by Xiang [30].”
Line 186-189
„The OA content measured in this study is in accordance with Qi’s results (30.66-32.71%) [20], but inconsistent with Weiss’s study (43.96-48.21%) [31]. OA content was stable in this study, which is contrary to the study by Lopez-Lopez et al., who claimed that OA content decreased as lactation progressed [32].”
Line 195-197
„Additionally, the percentage of EiA, EA, and NA decreased with the progression of lactation, which agrees with the results obtained by Nyuar [11].”
Line 207-210
„It has been demonstrated that the heart may be damaged by EA, and EA is undesirable for human consumption [33]. So, it is recommended that EA content in infant formulae be below 1% of the total fat content [34].”

Line 157
The title of Figure 1 „Gas chromatograms obtained from human milk” should be more informative.
Line 158-159
„The NA concentrations in human milk according to GC analysis are presented in Table 1. The NA concentration in all human milk samples decreased significantly (p < 0.05) during the first month of lactation.”
The value of the corelation cofficient (r=) should be given.
Line 160
The Authors reported the results as mg/g or mg/g fat?
„The average NA concentration on day 3 of lactation was about 5 times higher than that 160 on day 30 (p < 0.001), with values of 1.00 ± 0.24 mg/g and 0.18 ± 0.03 mg/g fat, respectively”
Was the total fat concentration determined in the analyzed milk? The results should be correlated with the total fat content. This would benefit and refine the obtained results (Table 1).

Discussion

Line 232
„The difference could be due to analytical methods, as well as differences between individuals from different locations.”
It is worth adding that also from which day of lactation the samples originated since there is a difference between the 3rd and the 6th day of lactation - both colostrum.
Line 235 - Table 3
It benefits the reader if the Authors will added additional citation to the Table.
 Sala-Vila A, Castellote AI, Rodriguez-Palmero M, Campoy C, López-Sabater MC. Lipid composition in human breast milk from Granada (Spain): changes during lactation. Nutrition. 2005 Apr;21(4):467-73.
Line 264 The „top” of the Figure 4 – Omega-9” is misleading, please correct.
Minor
Line 39 - Is „fatty acid (FAs)” – should be „fatty acids (FAs)”

Author Response

Response to Review Comments

(Manuscript ID nutrients-552980)

Reviewers' comments:

Reviewer #2: This paper nicely discusses the quantification of nervonic acid in huamn milk over the first month of lactation and compare with the level of different artificial milk. The manuscript covers the aim of the journal and the subject investigated is of worldwide interest. However, some points need to be addressed and the manuscript needs re-editing of some fragments.

The authors highly appreciate the reviewer’s comments, which the results and value of this work have been precisely summarized.

Overall

As stated by Authors, “eight healthy Chinese mothers were included to the study”, however, there are a lot of scientific reports which highlight the interindividual variation of different factors among lactating mothers, also by the authors themselves, so in my opinion, the obtained results, due to the low numer of lactating mothers, should be reported as preliminary and this should be reflected in the title of the manuscrypt.

More information has been added into the title of the manuscript.

Was the total fat concentration determined in the analyzed milk? For example, by using Miris Milk Analyser. The results should be correlated with the total fat content. This would benefit and refine the obtained results. Moreover, I have doubts whether the calculation of statistical differences among individual mothers (at the subsequent days of lactation) is justified (see Table 1 – last column, p value). Under the Table, the details concerning the statistical test used are missing. Moreover, the detailed data in Table 1 should be presented as supplementary file.

The human milk fat was extracted using classic Röse-Gottlieb method [1] and the fat contents were analyzed as our previous studies [2-6]. Follow the reviewer’s comments, we have added the fat content of eight samples during lactation day as one table in Supplementary file (Table S2). Table 2 has been moved to the supplementary file, as suggested (Table S3).

Line 46-47

“NA concentrations in colostrum have been found to range from 2- to 6-fold higher than those in mature milk [13,14].”

It benefits the reader if the Authors will added information “how mature milk”, namely weeks or months.

The information of the lactation day has been added as suggested. NA concentrations in colostrum (≤1 week postpartum) have been found to range from 2- to 6-fold higher than present in mature milk (>2 weeks and ≤16 weeks).

Line 70-79

2.2. Human milk samples

Some very important details concerning collections of milk are missing (i.e. using an electric breast pump, sampling of a single breast, etc.).

The detailed description is needed since there are a lot of data which point out the significant differences in human milk lipid fraction during single feeding (fat levels changing foremilk to hindmilk) as well as over the day.

The authors agree with the reviewer’s comments. The detailed information about the sample collection has been added (line 87-89).

Line 111

2.5. Quantitation of nervonic acid

“The NA concentration in human milk fat…” – the method used by Authors is semi-quantitative, so it would be more appropriate to use the word “level”.

We thank the reviewer’s comments. We have provided two formats of NA in human milk. One is the relative content which expressed as a percentage by weight of total fatty acids [7], which is easy to compare with other studies and provide more information of NA in infant formula production. Another is the absolute NA concentration. Here we used ‘concentration’ for a better understanding and distinguishing from the relative content.

Results

Some subsections contain the sentences which shoud be transferred to the Discussion. I suggest that the authors should modified these fragments:

Line 138-144

“NA occurs naturally in human milk; however, it has not been routinely reported in the global FA profile of human milk [24]. The main reason…”

Line 152-155

“The limits of detection and quantitation were lower than previous studies by Barros et al. [28] and Arcari et al. [8]. In general, …”

Line 164-166

“The results of the NA concentration in human milk during the three stages, that is, colostrum, transitional milk, and mature milk, are similar to those found by Xiang [30].”

Line 186-189

“The OA content measured in this study is in accordance with Qi’s results (30.66-32.71%) [20], but inconsistent with Weiss’s study (43.96-48.21%) [31]. OA content was stable in this study, which is contrary to the study by Lopez-Lopez et al., who claimed that OA content decreased as lactation progressed [32].”

Line 195-197

“Additionally, the percentage of EiA, EA, and NA decreased with the progression of lactation, which agrees with the results obtained by Nyuar [11].”

Line 207-210

“It has been demonstrated that the heart may be damaged by EA, and EA is undesirable for human consumption [33]. So, it is recommended that EA content in infant formulae be below 1% of the total fat content [8].”

The authors appreciate the reviewer’s comments. These fragments have been moved into the discussion where related.

Line 157

The title of Figure 1 “Gas chromatograms obtained from human milk” should be more informative.

Modified to as suggested.

Line 158-159

“The NA concentrations in human milk according to GC analysis are presented in Table 1. The NA concentration in all human milk samples decreased significantly (p < 0.05) during the first month of lactation.”

The value of the corelation cofficient (r=) should be given.

The authors thank review’s comments. The correlation coefficient (r=-0.822, p < 0.001) has been added into the manuscript (line 176).

Line 160

The Authors reported the results as mg/g or mg/g fat?

“The average NA concentration on day 3 of lactation was about 5 times higher than that 160 on day 30 (p < 0.001), with values of 1.00 ± 0.24 mg/g and 0.18 ± 0.03 mg/g fat, respectively”

Was the total fat concentration determined in the analyzed milk? The results should be correlated with the total fat content. This would benefit and refine the obtained results (Table 1).

The fatty acid in this study is expressed as mg/g milk fat. However, the reason we use mg/g milk fat instead of mg/mL is that we think this report format is more accurate. Although we have followed a widely used lipid extraction method and have repeated several times, the fat content analysis is not accurate, which is largely influenced by the experimental operation. Besides, fat content is the most variable constituent in human milk [9,10]. Therefore, we think the NA concentration presented as mg/g milk fat is better. As the response above, we have provided the fat content of all samples during lactation. Therefore, we can calculate the fatty acid concentrations using these data.

Discussion

Line 232

“The difference could be due to analytical methods, as well as differences between individuals from different locations.”

It is worth adding that also from which day of lactation the samples originated since there is a difference between the 3rd and the 6th day of lactation - both colostrum.

Added as suggested (line 251).

Line 235 - Table 3

It benefits the reader if the Authors will added additional citation to the Table.

Sala-Vila A, Castellote AI, Rodriguez-Palmero M, Campoy C, López-Sabater MC. Lipid composition in human breast milk from Granada (Spain): changes during lactation. Nutrition. 2005 Apr;21(4):467-73.

It is very nice work. The reference has been cited as suggested (reference 39)

Line 264 The “top” of the Figure 4 – Omega-9” is misleading, please correct.

We deleted the “top” in this Figure 4.

Minor

Line 39 - Is “fatty acid (FAs)” – should be “fatty acids (FAs)”

Corrected as suggested.

References

1.             Jensen, R.G. The Lipids of Human Milk; CRC Press: Florida, 1989.

2.             Qi, C.; Sun, J.; Xia, Y.; Yu, R.; Wei, W.; Xiang, J.; Jin, Q.; Xiao, H.; Wang, X. Fatty acid profile and the sn-2 position distribution in triacylglycerols of breast milk during different lactation stages. Journal of Agricultural and Food Chemistry 2018, 66, 3118-3126, doi:10.1021/acs.jafc.8b01085.

3.             Yuan, T.; Qi, C.; Dai, X.; Xia, Y.; Sun, C.; Sun, J.; Yu, R.; Zhou, Q.; Jin, Q.; Wei, W., et al. Triacylglycerol composition of breast milk during different lactation stages. Journal of Agricultural and Food Chemistry 2019, 67, 2272-2278, doi:10.1021/acs.jafc.8b06554.

4.             Sun, C.; Zou, X.; Yao, Y.; Jin, J.; Xia, Y.; Huang, J.; Jin, Q.; Wang, X. Evaluation of fatty acid composition in commercial infant formulas on the Chinese market: A comparative study based on fat source and stage. Int. Dairy J. 2016, 63, 42-51.

5.             Sun, C.; Wei, W.; Su, H.; Zou, X.; Wang, X. Evaluation of sn-2 fatty acid composition in commercial infant formulas on the Chinese market: A comparative study based on fat source and stage. Food Chemistry 2018, 242, 29-36, doi:https://doi.org/10.1016/j.foodchem.2017.09.005.

6.             Sun, C.; Wei, W.; Zou, X.; Huang, J.; Jin, Q.; Wang, X. Evaluation of triacylglycerol composition in commercial infant formulas on the Chinese market: A comparative study based on fat source and stage. Food Chemistry 2018, 252, 154-162, doi:https://doi.org/10.1016/j.foodchem.2018.01.072.

7.             Brenna, J.T.; Plourde, M.; Stark, K.D.; Jones, P.J.; Lin, Y.-H. Best practices for the design, laboratory analysis, and reporting of trials involving fatty acids. The American Journal of Clinical Nutrition 2018, 10.1093/ajcn/nqy089, nqy089-nqy089, doi:10.1093/ajcn/nqy089.

8.             29, U.-N. Validation on Compendial Procedures. The United States Pharmacopeia and National Formulary Rockville 2011, 749.

9.             Jensen, R.G. The lipids in human milk. Progress in Lipid Research 1996, 35, 53-92, doi:http://dx.doi.org/10.1016/0163-7827(95)00010-0.

10.           Wei, W.; Jin, Q.; Wang, X. Human milk fat substitutes: Past achievements and current trends. Progress in Lipid Research 2019, 74, 69-86, doi:https://doi.org/10.1016/j.plipres.2019.02.001.

Round 2

Reviewer 1 Report

The reviewer appreciates careful consideration of comments and revision on the manuscript. Two remaining items: 

Reference 16 is a review – if stating impaired white matter can cause cerebral palsy, reference the original article stating this.

Please include the use of 24:1n-9 FAME as the calibration standard and 13:0 as the internal standard.  

Author Response

Response to Review Comments

(Manuscript ID nutrients-552980)

The reviewer appreciates careful consideration of comments and revision on the manuscript. Two remaining items:

The authors thank the reviewer’s comments. We have addressed the reviewer’s comments from point by point as follows. The corresponding modifications are now included in the revised version of the manuscript.

Reference 16 is a review – if stating impaired white matter can cause cerebral palsy, reference the original article stating this.

The original article has been cited (ref 17). This part has been rewritten for better understanding (line 50-52).

Please include the use of 24:1n-9 FAME as the calibration standard and 13:0 as the internal standard.

The calibration lines of 24:1 n-9 (standard) and 23:0 (internal standard) are y = 9.47 106X (R2 = 0.9938) and y = 9.59 106X (R2 = 0.9965), respectively. This part has been added into the manuscript (line 163-164).
